# Performance of Serum Angiotensin-Converting Enzyme in Diagnosing Sarcoidosis and Predicting the Active Status of Sarcoidosis: A Meta-Analysis

**DOI:** 10.3390/biom12101400

**Published:** 2022-09-30

**Authors:** Xueru Hu, Li Zou, Shuyan Wang, Tingting Zeng, Ping Li, Yongchun Shen, Lei Chen

**Affiliations:** 1Department of Respiratory and Critical Care Medicine, West China Hospital of Sichuan University and Division of Pulmonary Diseases, State Key Laboratory of Biotherapy of China, Chengdu 610041, China; 2Department of Pediatric Surgery, West China Hospital of Sichuan University, Chengdu 610041, China

**Keywords:** angiotensin-converting enzyme, sarcoidosis, active status of sarcoidosis, diagnosis, meta-analysis

## Abstract

The usefulness of serum angiotensin-converting enzyme (sACE) for diagnosing sarcoidosis and determining the active status of sarcoidosis has been reported with varying outcomes. On the basis of the majority of published data, we conducted a meta-analysis to calculate the overall predictive accuracy of sACE in sarcoidosis disease and the active status of sarcoidosis. The inclusion of related research listed in Web of Science, PubMed, Scopus, and other literature databases was assessed. SROC curves were generated to characterize the overall test results after data on sensitivity, specificity, positive likelihood ratio (PLR), negative likelihood ratio (NLR), and diagnostic odds ratio (DOR) were combined. Publication bias was identified using Deeks’ funnel plot. Thirty-five publications with 8645 subjects met the inclusion criteria. The following are summary estimates of sACE diagnostic performance for sarcoidosis: sensitivity, 60% (95% confidence interval (CI), 52–68%); specificity, 93% (95% CI, 88–96%); PLR, 8.4 (95% CI, 5.3–13.3); NLR, 0.43 (95% CI, 0.36–0.52); and DOR, 19 (95% CI, 12–31). The area under the SROC curve (AUC) was 0.84 (95% CI, 0.80–0.87). Summary estimates for predicting the active status of sarcoidosis were as follows: sensitivity, 0.76 (95% CI, 0.61–0.87); specificity, 0.80 (95% CI, 0.64–0.90); PLR, 3.9 (95% CI, 2.1–7.3); NLR, 0.29 (95% CI, 0.17–0.49); and DOR, 13 (95% CI, 6–31). The AUC was 0.85 (95% CI, 0.82–0.88). There was no evidence of publication bias. Our meta-analysis suggests that measuring the sACE may assist in the diagnosis of sarcoidosis and predicting the active status of sarcoidosis, but the interpretation of the sACE results should be with caution. Future studies should validate our results.

## 1. Introduction

Sarcoidosis is a multisystemic disease that primarily affects the lungs (90% of the time) but can affect any organ in the body. Sarcoidosis is distinguished by the development of non-caseating epithelioid cell granulomas [1]. Sarcoidosis patients have a lower survival rate than the general population [2]. About 12% of all stage IV patients require ongoing oxygen therapy, and many experience numerous complications that ultimately result in irreversible lesions such as pulmonary fibrosis, cirrhosis, fatal arrhythmia, blindness, and other conditions that have a significant impact on these patients’ quality of life and lifespan [3]. It is difficult to make an accurate diagnosis of sarcoidosis due to the variety and non-specificity of its clinical characteristics and the diverse radiological presentation of its patients [4,5]. Sarcoidosis is currently diagnosed using a mix of clinical, radiological, and histological characteristics. To correctly diagnose sarcoidosis, additional etiologies in the differential diagnosis must also be eliminated [6]. One issue with identifying sarcoidosis is that biopsy is typically advised unless a patient has thoracic stage I (bilateral hilar lymphadenopathy unrelated to neoplastic or infectious disorders) or exhibits characteristic symptoms of Lofgren’s and Heerfordt’s syndromes [7,8,9]. Even before any treatment is started, the inflammatory activity of sarcoidosis should be assessed so that the response to treatment can be evaluated [9]. As a result, there is a need to search for reliable, less invasive approaches to diagnose sarcoidosis and assess the activity of the disease at same time.

Numerous serum indicators in sarcoidosis patients have been studied. Many of these biomarkers are assumed to play an important role in the development, resolution, or potentiation of the sarcoid granuloma or to be linked to the development of fibrosis and other disease sequelae or to provide protection against their occurrence [10]. Angiotensin-converting enzyme (ACE) is the most well-known serum biomarker in sarcoidosis; it has also been shown to be expressed as a membrane-bound protein in several human tissues, including the lungs, intestine, heart, and kidneys [11], and it has a crucial impact in the pathogenesis of severe acute respiratory syndrome coronavirus 2 infection [12]. Serum ACE (sACE) is an acid glycoprotein generated mostly by activated alveolar macrophages that convert angiotensin I to angiotensin II [13]. The level of sACE is commonly frequently used for laboratory test in patients with sarcoidosis and may correlate with disease activity, and it also reflects the total sarcoidosis granuloma burden [14,15,16]. However, whether sACE level can reliably be used to diagnose sarcoidosis and predict the active status of the disease remains controversial, as the results of existing research show high variability [17,18,19]. The predictive value of sACE still needs to be further verified by the methods of systematic review and meta-analysis, which is essential to prove the generalization of predictive performance. This research aimed to summarize the usefulness of sACE for diagnosing sarcoidosis and determining the active status of sarcoidosis. 

## 2. Materials and Methods

The Cochrane Diagnostic Test Accuracy Working Group’s suggested procedures and the Preferred Reporting Items for Meta-analysis statement’s guidelines were followed in the conduct of this study [20]. This retrospective meta-analysis did not require approval from the institutional review board.

### 2.1. Search Strategy

The diagnostic performance of sACE for sarcoidosis or active sarcoidosis was assessed in original articles published up to August 2022. To find these articles, Web of Science, PubMed, OVID, and Scopus were searched. “sarcoidosis OR active sarcoidosis” AND “angiotensin-converting enzyme OR ACE” AND “sensitivity” OR “specificity” OR “accuracy” made up the search term. Only human and clinical studies were included in the search results that qualified. Using PubMed’s “related articles” feature, other papers were also discovered. We manually searched recognized article references to locate other publications.

### 2.2. Selection of Publications

The full-text versions of any research that could not be quickly ruled out were retrieved after we had examined the titles and abstracts of the identified publications. Finally, publications that met the following requirements were included in current meta-analysis: (1) sACE was used to diagnose sarcoidosis or determine the active status of sarcoidosis; (2) sufficient data were included to calculate the true positive (TP), false positive (FP), false negative (FN), and true negative (TN) values of sACE; and (3) studies contained original research published in English. Studies with less than 20 subjects were disregarded to prevent selection bias. Reviews, editorials, case reports, and conference abstracts were removed.

### 2.3. Data Extraction and Quality Assessment

Two authors (X.H. and L.Z.) evaluated each publication’s eligibility independently and took the following data from each one: the names of first author, the year of publication, the country, the number of cases and controls, the diagnostic standard, diseased region, the method, the study design, the cutoff, TP, FP, FN, TN, and values. Data extraction disagreements were settled by consensus. When information was not reported in the article, an effort was made to get in touch with the writers. Only the data linked to the highest diagnostic accuracy were included in current meta-analysis for studies that examined a variety of cutoff values.

To evaluate the methodological quality of each study, the Quality Assessment of Diagnostic Accuracy Studies (QUADAS)-2 tool was used. The risk of bias was analyzed across all four categories, comprising the four domains of patient selection, index test, reference standard, and flow and timing with the first three focusing on applicability [21].

### 2.4. Statistical Analysis

The meta-analysis was performed using the suggested standard methodologies for diagnostic precision [22]. We determined the sensitivity, specificity, positive likelihood ratio (PLR), negative likelihood ratio (NLR), and diagnostic odds ratio (DOR), along with the accompanying 95% confidence intervals (CIs) for each research in order to examine the test accuracy. Additionally calculated were the summary receiver operating characteristic (SROC) curves and area under the SROC curve (AUC) [23].

Heterogeneity among studies was calculated using the chi-squared test and Fisher’s exact test [24]. If significant heterogeneity existed among studies, subgroup and meta-regression analyses were conducted using covariates reported in most included studies, as follows: cutoff values, study design (prospective vs. retrospective or not reported), sample size (<100 subjects vs. ≥100 subjects), sampling method (consecutive vs. random or not reported), measurement method (spectrophotometric vs. other), diseased region (ocular sarcoidosis vs. other), and ethnicity (Caucasian vs. Asian). Publication bias was identified using Deeks’ funnel plot [25]. Fagan nomograms were used to compute the posttest probability (PTP) using an overall prevalence of 20%. In this meta-analysis, STATA 15.0 (Stata Corp., College Station, TX, USA) and RevMan 5.2 statistical tools were used (Cochrane Collaboration, Oxford, UK). The threshold for statistical significance was set at *p* 0.05 for all two-sided statistical tests. 

## 3. Results

### 3.1. Study Selection

We located 995 records during the literature search. Of these, we identified 64 relevant studies and retrieved their full-text versions. We also retrieved the full texts of six articles obtained during a manual bibliography search. Then, during the second screening stage, we selected 35 studies [15,17,19,26,27,28,29,30,31,32,33,34,35,36,37,38,39,40,41,42,43,44,45,46,47,48,49,50,51,52,53,54,55,56,57] satisfying the eligibility criteria with participants and analyzed their data for our review (Appendix A). 

### 3.2. Clinical Characteristics of the Included Studies

In total, we assessed data from 8645 patients for the utility of sACE in diagnosing sarcoidosis or predicting the active status of the disease. The sample size of included studies varied from 23 to 1541 participants. Among the 35 included studies, 31 [17,26,27,28,29,30,31,32,33,35,36,37,38,39,40,41,42,43,44,45,46,47,48,49,50,51,52,53,54,55,56] with 7571 subjects used sACE to diagnose sarcoidosis, while 9 studies [15,19,26,29,32,34,35,45,57] with 1074 subjects evaluated the predictive accuracy of the active status of sarcoidosis. The clinical summaries of included studies are listed in Table 1 and Table 2.

### 3.3. Risk of Bias Assessment

Figure 1 shows the risk of bias across various domains as per the results of applying the QUADAS-2 tool. In this meta-analysis, only 1 out of the 35 studies had a high risk of patient selection bias [26], and two studies had high risks of bias in patient flow and timing [26,41]. 

### 3.4. Diagnosis Utility of sACE for Sarcoidosis

Thirty-one studies reported the utility of sACE for diagnosing sarcoidosis (Table 1). The pooled sensitivity and specificity of sACE for patients with sarcoidosis were 60% (95% CI, 52–68%) and 93% (95% CI, 88–96%), respectively (Figure 2). Additionally, the DOR was 19 (95% CI, 12–31), the PLR was 8.4 (95% CI, 5.3–13.3), the NLR was 0.43 (95% CI, 0.36–0.52), and the AUC was 0.84 (95% CI, 0.80–0.87) (Figure 3). Fagan’s nomogram (Appendix A) showed a moderate clinical utility of sACE for diagnosis (positive, 68%; negative, 10%), which was significantly different from the pretest probability 20%. 

We found significant between-study variability (heterogeneity) with a chi-squared *p* value <0.001 and I^2^ values of 89.33% and 95.60%. In additional, we conducted a meta-regression and subgroup analysis on the basis of cutoff values, study design (prospective vs. retrospective or not reported), sample size (<100 subjects vs. ≥100 subjects), sampling method (consecutive vs. random or not reported), measurement method (spectrophotometric vs. other), diseased region (ocular sarcoidosis vs. other), and ethnicity (Caucasian vs. Asian). The outcome showed that sampling method had an influence on the results of sensitivity and specificity, and the measurement method affected the specificity of sACE (Appendix A). The publication bias in the final set of studies was evaluated using Deeks’ funnel plot asymmetry test. The slope coefficient had a value of 0.44, indicating the absence of publication bias (Appendix A). 

Among the included studies, there were eight on the diagnosis of ocular sarcoidosis (Appendix A). In this subgroup, the results of interest were as follows: sensitivity, 0.48% (95% CI, 35–61%), and specificity, 96% (95% CI, 89–99%) (Appendix A). Additionally, the AUC was 0.77 (95% CI, 0.73–0.80) (Appendix A). Deeks’ funnel plot asymmetry test showed *p = 0.99*, suggesting that there was no obvious asymmetry. All of the results are summarized in Table 3. Regarding subgroup analysis by ethnicity, summary estimates for predicting the diagnosis performance of sarcoidosis in Caucasians and Asians are shown in Appendix A.

### 3.5. Predictive Accuracy of sACE for the Active Status of Sarcoidosis

Nine studies were related to the activity of sarcoidosis in this meta-analysis (Table 2). For heterogeneity examination, the I^2^ values of sensitivity and specificity were 92.09% (95% CI, 88.29–95.89%) and 86.24% (95% CI, 78.50–93.98%), respectively, and a random-effects approach was used to pool the data. The sACE had medium predictive performance based on the pooled sensitivity of 0.76 (95% CI, 0.61–0.87), specificity of 0.80 (95% CI, 0.64–0.90) (Figure 4), PLR of 3.9 (95% CI, 2.1–7.3), NLR of 0.29 (95% CI, 0.17–0.49), and DOR of 13 (95% CI, 6–31). The AUC was 0.85 (95% CI, 0.82–0.88) (Figure 5). The slope coefficients for the ACE were associated with a *p*-value of 0.1, suggesting a low likelihood of such bias. Because of the limited number of studies that were reviewed for this analysis, we did not carry out a meta-regression to determine the potential source of heterogeneity.

## 4. Discussion

Sarcoidosis is a systemic granulomatous disease of unknown origin featured by a wide variety of clinical presentations [58]. The disease occurs in adults aged < 40 years, peaking in 20–29-year-olds, with a second peak occurring in patients > 50 years of age, especially in women [59]. Sarcoidosis is an exclusionary diagnosis without objective diagnostic criteria. Confirming the diagnosis of sarcoidosis is a major clinical problem in daily practice. Identification of a validated biological marker would be helpful for diagnosing sarcoidosis. In 1975, Lieberman et al. [26] discovered that sACE activity was increased in the blood of patients with sarcoidosis. Growing studies show that sACE may have potential role both in the diagnosis of sarcoidosis and in predicting the active status of patients with sarcoidosis, but results vary between different studies. Thus, we carried out a meta-analysis to summarize the performance of sACE in diagnosing sarcoidosis and predicting the active status of sarcoidosis. 

The meta-analysis of sACE diagnostic performance revealed pooled values for sensitivity (0.60) and specificity (0.93). The AUC of 0.84 was obtained using the SROC curve, which provides a global summary of test results and displays the tradeoff between sensitivity and specificity, suggesting a good overall accuracy. The DOR value combines sensitivity and specificity data into a single number ranging from 0 to infinity, indicating greater discriminatory test performance. In this meta-analysis, the mean DOR was 19, showing that sACE appeared to be effective in identifying sarcoidosis. Both PLR and NLR were classical diagnostic accuracy indicators, and likelihood ratios greater than 10 and less than 0.1 were regarded as strong markers for ruling in or ruling out a diagnosis, respectively [60]. The obtained PLR value of 8.4 suggests that a patient has about a nine-fold higher chance of being diagnosed with sarcoidosis compared to the control, but this value was not high enough for clinical utility. In addition, the NLR was 0.43, which suggests that if the result of sACE is negative, the probability that the patient has sarcoidosis is 43%, and it is not low enough to rule out a diagnosis of sarcoidosis. Sarcoidosis also creates a sight-threatening intraocular inflammatory syndrome that is regarded as a prominent clinical entity of uveitis around the world [61]. Because of the possibility for visual impairment, intraocular tissue biopsies are rarely conducted. The subgroup analysis indicates a sensitivity of 35–61% and a specificity of 89–99% for sACE level in the diagnosis of ocular sarcoidosis. These findings are similar to previous reports (sensitivity, 22–73%; specificity, 70–99.5%) [17,44,53,62,63]. In comparison to sACE, most studies indicated that serum-soluble interleukin-2 receptor is more helpful in the diagnosis of ocular sarcoidosis, as it demonstrates high sensitivity and specificity [17,50,53,54]. According to the results, the clinical meaning of sACE assay is that for patients who have the most common clinical features and imaging changes with the suspicious of sarcoidosis, elevated levels of sACE can be used as a supplementary diagnostic test for sarcoidosis and instruct whether to perform more invasive operations in clinical practice. 

On the basis of this meta-analysis, sACE was found to not be sufficiently sensitive and specific enough to diagnose sarcoidosis alone. One study used serum chitotriosidase and sACE values with the multiplication of these values by each other to further improve the diagnostic accuracy. The combination had 90.5% sensitivity, 79.3% specificity, and 90.5% positive and 79.3% negative predictive values, respectively [51]. Ma et al. [64] found that a random forest model had adequate performance for distinguishing between sarcoidosis and tuberculosis by incorporating statistically significant diagnostic factors. In their study, the AUC of the random forest prediction model was 0.915. Different combination models of biomarkers and a new diagnosis model need to be verified further in the future to improve the diagnostic accuracy of sACE.

Rational analysis of the ACE results is necessary. Firstly, the reason why is that the levels of ACE in plasma are stable in a given individual but vary greatly between subjects [65]. A normal range of ACE levels is one that varies three-fold in the tested population from the mean value, even up to 5.7 times among subjects [65,66]. The sACE levels that are within the normal range may actually be high in some people, which is related to the various polymorphisms of the ACE gene. Homozygous carriers of D (DD) or I (II) express the highest and lowest ACE levels, respectively, with intermediate ACE levels for heterozygous ID individuals [67,68]. ACE levels are 30% and 60% greater in individuals with one or two D alleles than in individuals with the II genotype, respectively [68]. Secondly, it has been reported that the ACE I/D polymorphism is associated with susceptibility to sarcoidosis in European and East Asian populations [69]. The mean sACE activity in Asian and Caucasian individuals is significantly higher in those with the DD genotype than in those with the II genotype, with the ID genotype resulting in intermediate values [70]. The I-allele is more common in Asian populations than in Caucasian populations [13]. Recently, use of new genotype-specific reference values for sACE levels has been reported [71,72]. When sACE values calculated diagnostically were compared to the appropriate genotype-specific reference range, the sensitivity and specificity for diagnosing sarcoidosis were 65–70% and 58%, respectively, compared to 47–57% and 77% when utilizing a reference range unsegregated by genotype [72]. The new method improved the sensitivity of tests for sACE during disease follow-up but decreased the specificity of sACE in sarcoidosis. In clinical practice, a Z-score has been developed that corrects the ACE activity for the I/D polymorphism [19,20]. Using the correction for this I/D polymorphism results in a different interpretation in 8.5% of measurements [73]. The fact is that only 20% of the entire variance in tissue and serum ACE is accounted for by this specific ACE I/D polymorphism [68,74]. Thirdly, sACE levels increase in only 60% of sarcoidosis patients [74,75,76], and a large percentage of people with granulomas in the lung will not have high sACE activity. Moreover, sACE levels have been found to be increased in several other inflammatory diseases such as tuberculosis, berylliosis, histoplasmosis, and Gaucher’s disease [28,77,78,79]. Thus, if an individual is below average in sACE level, even though sarcoidosis increases a person’s blood ACE level, it may not be diagnosed with the disease on the basis of blood ACE activity. 

There are other factors influencing sACE activity that have been reported, such as age, gender, cigarette smoking, and ACE inhibitor (ACEI) [80]. Lieberman et al. [26] found that enzyme activity was greater in children than in adults. Recently, a new analysis considering age distribution in 1269 patients with sarcoidosis reported a significant decrease in sACE level among patients ≥ 60 years of age [81]. The difference in the results may be affected by the varying age composition of the population between studies. Moreover, Lieberman et al. [26] showed that enzyme activity was greater in male control subjects than in female control subjects of comparable age. However, most studies showed that sACE level did not differ according to sex during a subgroup analysis of sarcoidosis patients [81,82,83]. Cigarette smoking alters the quantity, type, and functional activity of lung immune and inflammatory cells and is a major contributor to the development of sarcoidosis [84]. Cigarette smoking is not only associated with an increase in sACE activity in sarcoidosis patients, but it also increases the concentration of ACE in alveolar fluid with tobacco use [85,86]. The data show alveolar macrophages in smokers exhibit more ACE activity than those in non-smokers, and the activity is even more elevated in sarcoidosis patients [87]. Compared to alveolar macrophages from non-smokers, the ACE activities in those from smokers and patients with sarcoidosis are, respectively, three and five times greater (*p* < 0.01) [88]. Thus, the interpretation of sACE results should be combined with history taking about cigarette smoking. In addition, ACEI have an effect on the level of sACE; if patients with sarcoidosis use ACEI, sACE levels cannot be used in diagnosis or disease monitoring, and should be interpreted carefully [89].

Importantly, determining whether a patient has active disease or is in remission is a significant difficulty in the management of sarcoidosis patients [19]. The data show the pooled sensitivity and specificity values were 0.76 and 0.80 for sACE to predict sarcoidosis activity, respectively, and the AUC was 0.85. The overall results suggested that the overall predictive ability of sACE for active sarcoidosis exists in the middle of the spectrum. The results show that sACE is helpful in determining the active status of sarcoidosis, and detection of the decrease in elevated blood ACE is a good marker for efficiency of therapy. 

The level of sACE was correlated with the granuloma burden in sarcoidosis, which may be a useful adjunctive measure to assess the clinical course of the disease [90]. However, the greater overlapping of sACE activity between sarcoidosis patients and healthy subjects may limit the value of this parameter with regard to monitoring disease activity, especially in patients with chronic disease [91]. Although the data were not that sound, we believe the examination of sACE is helpful for clinicians to make a treatment plan for such patients to some extent.

Corticosteroids are the preferred medication for the treatment of sarcoidosis that work by inhibiting pro-inflammatory cytokines and chemokines implicated in cell-mediated immune responses and granuloma formation [92,93]. Corticosteroid treatment reflects the sACE level in patients with sarcoidosis [19]. It has been reported that resolution of the sarcoidosis disease state or therapeutic control with adequate doses of corticosteroids brought elevated sACE levels down into the normal range [26,94]. When corticosteroid medication is tapered, the enzyme level may re-elevate to basal levels even if there is no disease relapse if the raised enzyme level is much lower than the pretreatment level [95]. Thus, a history of corticosteroids use should be taken into consideration if sACE was used to evaluate the activity of disease.

The limitations of this meta-analysis should be pointed out in order to properly evaluate its results. Although our stringent inclusion and exclusion criteria may have reduced selection bias, they also resulted in a limited final collection of trials, for which the statistical power may not be sufficient to draw firm conclusions about the diagnostic and evaluative capabilities of sACE. Significant heterogeneity among the included studies was identified, and sampling and measuring techniques all affected the outcomes of sensitivity or specificity. Future studies should aim for greater rigor to decrease the risk of bias. Last but not least, we only used papers from a restricted number of literature databases that were published in English. The deletion of unpublished studies, studies published in other languages, and research published in journals not indexed in the databases we examined may have influenced our findings.

## 5. Conclusions

Taken together, this meta-analysis suggests that assaying sACE of the interpretation should be with cautious. sACE may be useful as an effective index to assess the active status of the disease and to guide the comprehensive management of patients with sarcoidosis.

## Figures and Tables

**Figure 1 biomolecules-12-01400-f001:**
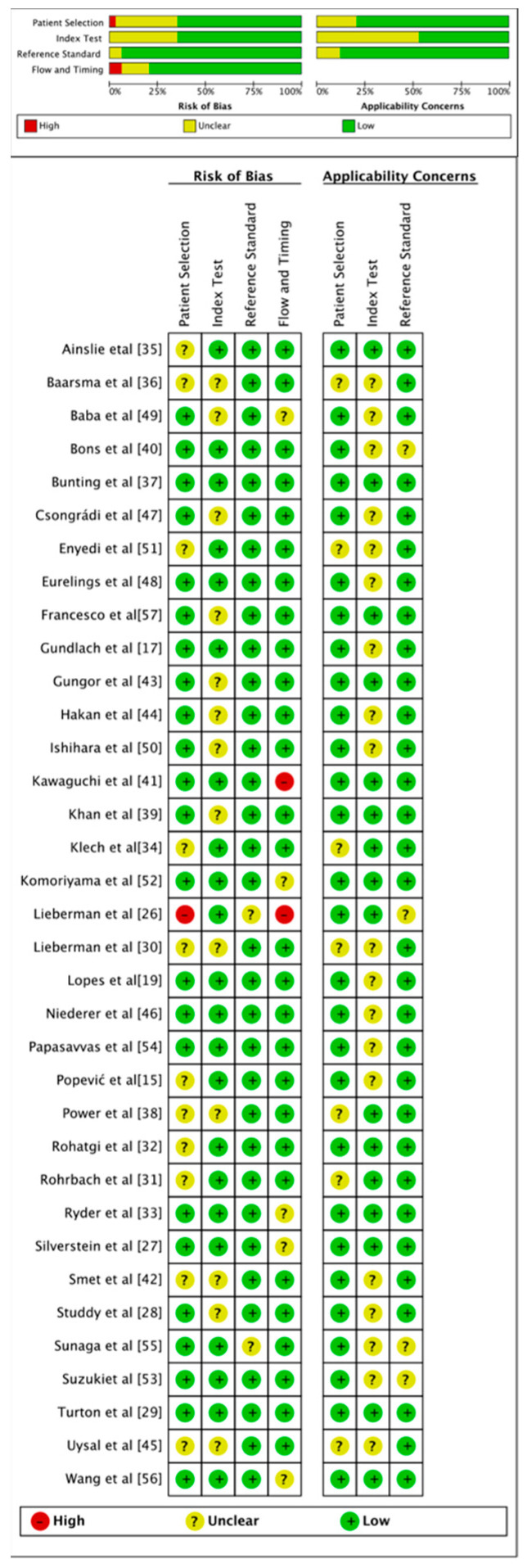
Quality assessment of studies included in the meta-analysis.

**Figure 2 biomolecules-12-01400-f002:**
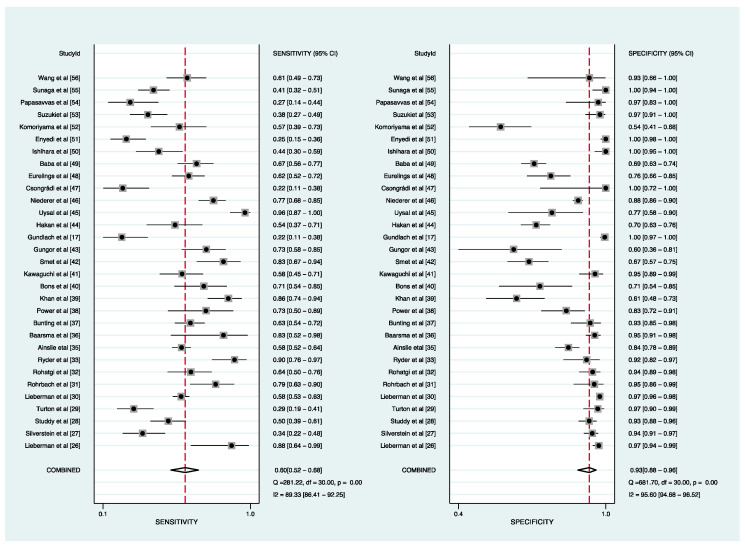
Forest plots of sensitivity and specificity of the sACE for diagnosing sarcoidosis. **Left**: sensitivity; **Right**: specificity.

**Figure 3 biomolecules-12-01400-f003:**
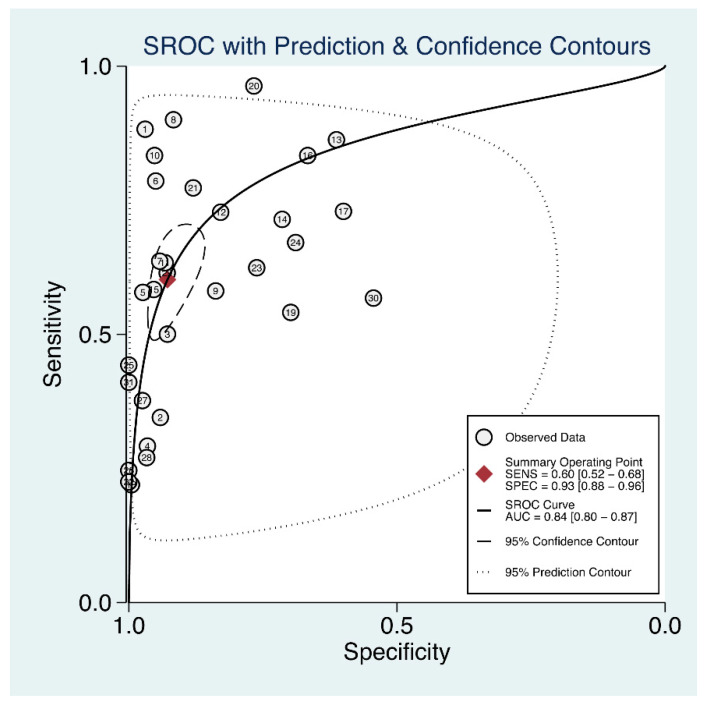
Summary receiver operating characteristic curve.

**Figure 4 biomolecules-12-01400-f004:**
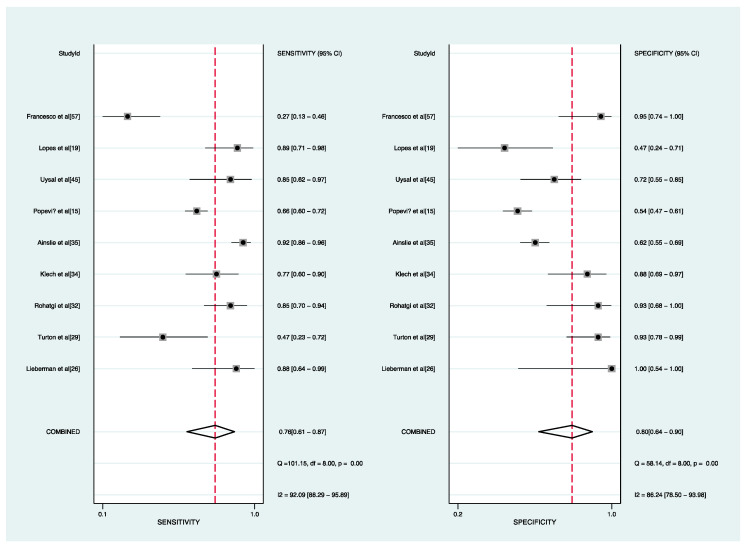
Forest plots of sensitivity and specificity of the sACE for diagnosing active sarcoidosis **Left**: sensitivity; **right**: specificity.

**Figure 5 biomolecules-12-01400-f005:**
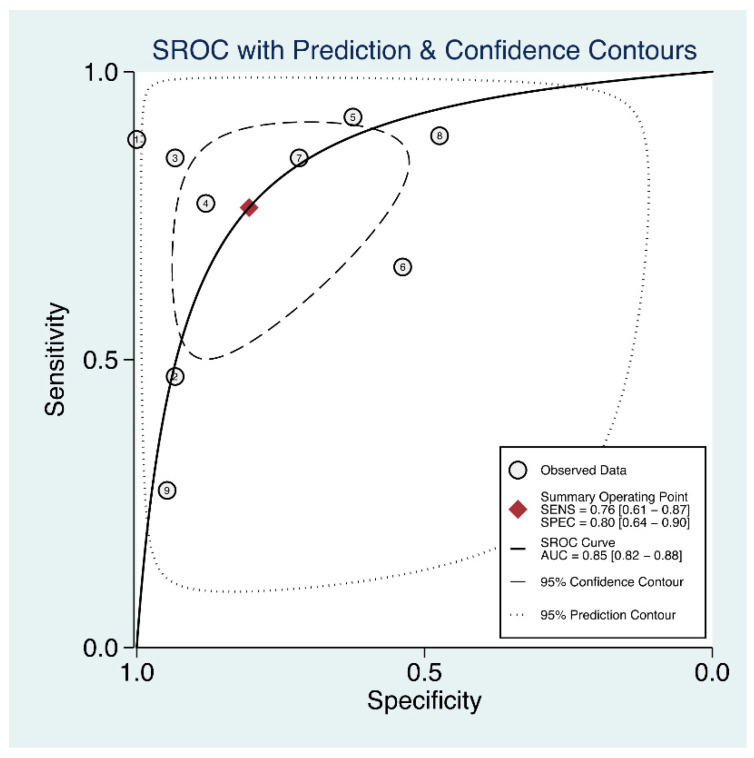
Receiver operating characteristic curves showing the performance of the sACE for diagnosing active sarcoidosis.

**Table 1 biomolecules-12-01400-t001:** The clinical summary of included studies for sarcoidosis.

Author	Year	Country	Criterial	Diseased Region	Method	Sarcoidosis					Control					Design	Cut-Off	TP	FP	FN	TN
						Case	Age	Gender (M/F)	sACE	Unit	Control	Age	Gender (M/F)	sACE	Unit						
Lieberman et al. [26]	1975	US	Histology	NA	Spectrophotometry	17	NA	NA	13.65 ± 2.26	U/mL	296	NA	NA	NA	U/mL	NA	11.6	15	9	2	287
Silverstein et al. [27]	1976	US	NA	NA	Spectrophotometry	58	48.3 ± 4.8	NA	48.3 + 4.8	nmol/min/mL	256	NA	NA	NA	nmol/min/mL	NA	30.48	20	15	38	241
Studdy et al. [28]	1978	UK	Histology	NA	Spectrophotometry	90	NA	NA	58 ± 24	nmol/min/mL	194	NA	NA	NA	nmol/min/mL	NA	52	45	14	45	180
Turton et al. [29]	1979	UK	Histology	NA	Spectrophotometry	72	37(19–67)	31/41	35.6 ± 14.4	U/mL	87	NA	NA	NA	U/mL	NA	41	21	3	51	84
Lieberman et al. [30]	1979	US	NA	NA	Spectrophotometry	391	39.6 ± 10	NA	NA	U/mL	1150	NA	NA	NA	U/mL	NA	35	226	30	165	1120
Rohrbach et al. [31]	1979	US	Histology	NA	Radiochemistry	42	44.8	22/20	71.9 ± 19.2	U/mL	60	NA	NA	40.2 ± 8.6	U/mL	R	57	33	3	9	57
Rohatgi et al. [32]	1980	US	Histology	NA	Radiochemistry	55	17–69	NA	NA	mU/mL	122	NA	NA	NA	mU/mL	NA	137.5	35	7	20	115
Ryder et al. [33]	1981	US	Clinical findings and/or histology	NA	Spectrophotometry	40	NA	NA	44 ± 8	kU/L	60	NA	NA	NA	kU/L	P	35	36	5	4	55
Ainslie et al. [35]	1985	South Africa	Histology	NA	Spectrophotometry	303	NA	NA	NA	nmol/min/mL	210	NA	NA	NA	nmol/min/mL	R	54	176	34	127	176
Baarsma et al. [36]	1987	Netherlands	Clinical and biochemical findings	Ocular sarcoidosis	Spectrophotometry	12	NA	NA	NA	U/L	209	NA	NA	NA	U/L	NA	50	10	10	2	199
Bunting et al. [37]	1987	Canada	Clinical findings or histology	NA	Fluorimetry	120	NA	NA	NA	U/L	74	NA	NA	NA	U/L	NA	50	76	5	44	69
Power et al. [38]	1995	US	Histology	Ocular sarcoidosis	Spectrophotometry	22	35 (22–64)	10/12	NA	U/L	70	39 (17–58)	29/41	NA	U/L	R	52	16	12	6	58
Khan et al. [39]	1998	Pakistan	Clinical findings and/or histology	NA	Spectrophotometry	51	NA	NA	104.44	IU/L	62	NA	NA	NA	IU/L	R	52	44	24	7	38
Bons et al. [40]	2007	Netherlands	WASOG	NA	Spectrophotometry	35	45.67 ± 8.8	18/17	22.0 (16.0–24.0)	U/L	35	47.37 ± 9.8	16/19	13.5 (12.0–17.8)	U/L	p	16.5	25	10	10	25
Kawaguchi et al. [41]	2007	Japan	Histology	Ocular sarcoidosis	NA	60	NA	NA	NA	NA	86	NA	NA	NA	NA	R	NA	35	4	25	82
Smet et al. [42]	2010	Belgium	Histology	NA	Spectrophotometry	36	NA	21/15	48 (39–80)	U/L	117	NA	65/52	27 (19–40)	U/L	R	34	30	39	6	78
Gungor et al. [43]	2015	Turkey	Histology	NA	Spectrophotometry	48	44.29 ± 10.90	9/39	49.50 ± 41.04	U/L	20	38.05 ± 8.36	10/10	25.44 ± 13.38	U/L	P	NA	35	8	13	12
Gundlach et al. [17]	2016	Germany	IWOS	Ocular sarcoidosis	Spectrophotometry	41	NA	NA	NA	U/mL	220	NA	NA	NA	U/mL	R	82	9	1	32	219
Hakan et al. [44]	2017	Netherlands	IWOS	Ocular sarcoidosis	Spectrophotometry	37	NA	NA	NA	U/L	212	NA	NA	NA	U/L	R	51	20	64	17	148
Uysal et al. [45]	2018	Turkey	WASOG	NA	ELISA	59	NA	NA	NA	mg/mL	25	46.1 ± 8.4	12/13	NA	mg/mL	NA	5.37	52	7	2	23
Niederer et al. [46]	2018	UK	IWOS	NA	NA	110	NA	NA	32 (21–47)	NA	925	NA	NA	NA	NA	P	52	85	111	25	814
Csongrádi et al. [47]	2018	Hungary	Histology	NA	Fluorimetry	40	39.0 ± 11.0	22/18	NA	U/L	11	54.0 ± 15.2	8/3	9.12 ± 2.07	U/L	P	21.4	9	0	31	11
Eurelings et al. [48]	2019	Netherlands	ATS/ERS/WASOG	NA	Spectrophotometry	101	43 (35–52)	50/51	77 (44–109)	U/mL	88	44 (31–57)	36/52	51(31–69)	U/mL	R	68	63	21	38	67
Baba et al. [49]	2019	Japan	JSSOG 2015	NA	Spectrophotometry	79	65.0 (55.0–71.0)	27/52	20.3 (16.0–24.4)	IU/L	299	NA	NA	15.4 (12.8–18.5)	IU/L	R	17.7	53	93	26	206
Ishihara et al. [50]	2020	Japan	Clinical findings and/or histology	Ocular sarcoidosis	Spectrophotometry	52	58.8 ± 15.2	13/39	26.4 ± 9.18	U/L	74	49.6 ± 16.5	40/34	13.5 ± 3.83	U/L	R	NA	23	0	29	74
Enyedi et al. [51]	2020	Hungary	Histology	NA	Fluorimetry	69	40.9 (±12.3)	30/39	11.89 (10.5–13.7)	U/L	168	NA	NA	NA	U/L	P	NA	17	0	52	168
Komoriyama et al. [52]	2020	Japan	Histology	NA	Spectrophotometry	37	57 ± 12	7/30	15.5 ± 7.9	IU/L	57	61 ± 9	12/45	12.4 ± 6.9	IU/L	R	13.5	21	26	16	31
Suzukiet al. [53]	2021	Japan	Histology	Ocular sarcoidosis	ELISA	77	54.6 ± 14.8	NA	NA	U/L	79	43.2 ± 17.8	NA	NA	U/L	R	12.7	29	2	48	77
Papasavvas et al. [54]	2021	Switzerland	IWOS	Ocular sarcoidosis	Spectrophotometry	37	54.52 ± 23.74	NA	49.17 ± 29	U/L	30	41 ± 11.3	NA	27.4 ± 15.34	U/L	R	NA	10	1	27	29
Sunaga et al. [55]	2022	Japan	Histology	NA	Spectrophotometry	112	61(14–86)	36/78	20.2 (0–56.4)	IU/L	62	NA	NA	NA	IU/L	R	21.4	46	0	66	62
Wang et al. [56]	2022	China	Histology	NA	Spectrophotometry	70	55.94 ± 11:83	14/56	56:61 ± 30.80	U/L	14	55.25 ± 16:70	8/6	28:07 ± 14.11	U/L	P	44	43	1	27	13

FN, false negative; FP, false positive; TN, true negative; TP, true positive; NA: not available; P: prospective; R: retrospective WASOG: World Association of Sarcoidosis and Other Granulomatous Disorders; ATS: the American Thoracic Society; ERS: the European Respiratory Society; IWOS: International Workshop on Ocular Sarcoidosis; sACE: serum angiotensin-converting enzyme.

**Table 2 biomolecules-12-01400-t002:** The clinical summary of included studies for active status of sarcoidosis.

Author	Year	Country	Criterial	Method	Sarcoidosis					Control					Design	Cut-Off	TP	FP	FN	TN
					Number	Age	Gender (M/F)	sACE	Unit	Number	Age	Gender (M/F)	sACE	Unit						
Lieberman et al. [26]	1975	US	Histology	Spectrophotometry	17	NA	NA	13.65 ± 2.26	U/mL	6	NA	NA	NA	U/mL	NA	11.6	15	0	2	6
Turton et al. [29]	1979	UK	Histology	Spectrophotometry	17	NA	NA	NA	U/mL	30	NA	NA	NA	U/mL	NA	41	8	2	9	28
Rohatgi et al. [32]	1980	UK	Histology	Radiochemistry	40	NA	NA	177.7 ± 59.7	U/mL	15	NA	NA	NA	U/mL	NA	137.5	34	1	6	14
Klech et al. [34]	1982	Austria	Histology	Spectrophotometry	35	NA	NA	NA	U/mL	25	NA	NA	NA	U/mL	NA	24	27	3	8	22
Ainslie et al. [35]	1985	South Africa	Histology	Spectrophotometry	114	NA	NA	100.4 ± 45.7	nmol/min/mL	189	NA	NA	51.9 ± 19.9	nmol/min/mL	R	54	105	71	9	118
Popević et al. [15]	2016	Serbia	Histology	Spectrophotometry	230	NA	NA	43 (26–62)	U/L	199	NA	NA	30 (20–44)	U/L	NA	32	152	92	78	107
Uysal et al. [45]	2018	Turkey	WASOG	ELISA	20	45.15 ± 11.55	4/16	NA	U/L	39	42.4 ± 10.4	15/24	NA	U/L	NA	26.23	17	11	3	28
Lopes et al. [19]	2019	Brazil	WASOG	ELISA	27	46 (13.5)	8/19	470.96 (407.81–534.09)	ng/mL	19	56 (18.5)	7/12	337.866 (262.47–413.26)	ng/mL	NA	270	24	10	3	9
Francesco et al. [57]	2021	Italy	Histology	ELISA	33	63(56.5–74.5)	5/28	34.0 (26.0–62.0)	UI/L	19	65.0 (56.0–71.0)	NA	NA	UI/L	NA	65	9	1	24	18

FN, false negative; FP, false positive; TN, true negative; TP, true positive; NA: not available; P: prospective; R: retrospective; sACE: serum angiotensin-converting enzyme; WASOG: World Association of Sarcoidosis and Other Granulomatous Disorders.

**Table 3 biomolecules-12-01400-t003:** The diagnosis performance of sACE by status of sarcoidosis.

	Sensitivity	Specificity	PLR	NLR	DOR	Area under the Curve (95% CI)
Sarcoidosis	60%	93%	8.4	0.43	19	0.84 (0.80–0.87)
Ocular sarcoidosis	48%	96%	13.3	0.54	25	0.77 (0.73–0.80)
Active sarcoidosis	76%	80%	3.9	0.29	13	0.85 (0.82–0.88)

sACE: serum angiotensin-converting enzyme; PLR: positive likelihood ratio; NLR: negative likelihood ratio; DOR: diagnostic odds ratio; CI: confidence interval.

## Data Availability

Not applicable.

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
