# Peer review of "Performance of Serum Angiotensin-Converting Enzyme in Diagnosing Sarcoidosis and Predicting the Active Status of Sarcoidosis: A Meta-Analysis"

_biomolecules, 2022, doi:10.3390/biom12101400_

Round 1

Reviewer 1 Report

Frankly speaking,  since 1975,  when J. Liberman found that blood ACE activity  in a group of sarcoidosis patients increased,  dozen (if not hundred) papers were published reproducing these results.

What this paper (Hu et al. 2022) add novel to this field?

To my opinion this paper did not explain a physical sense of the phenomenon-elevation of blood ACE in sarcoidosis .

1)    ACE activity vary 3-4 fold in population (Alnhenc-Gelas, 1991, Samokhodskaya, 2021) and thus blood ACE measurement has limited diagnostic value for given individual, because if genetically determined ACE expression(not only ACE I/D polymorphism, but other polymorphic variants as well (Danser, 2007)  in a given individual is low (60% of mean) even if sarcoidosis in a given individual  will lead to 2-fold increase of blood ACE, this patient with 120% of blood ACE (from mean of population) will not be considered as having sarcoidosis based on blood ACE activity.

2)    Another issue: only 60% of patients with histologically proven lung sarcoidosis has an elevated blood ACE -discussed in  (Romer, 1984, Danilov, 2022).

Therefore,  if patient has 200% of blood ACE -it is likely candidate for sarcoidosis, but significant proportion of patients with granulomas in the lung only will have no elevated ACE activity.

Thus,  measurement of blood ACE (detection of the decrease in elevated blood ACE ) is a good marker for efficiency of therapy , but measurement of simply ACE activity in the blood has very limited value -61% (as you mentioned) for personalized medicine

Such simple explanations (for common readers) are absent in the proposed manuscript

Reviewer 2 Report

The authors conducted a meta-analysis to evaluate the diagnostic value of serum ACE (sACE) in patients with sarcoidosis. While sACE has been widely used for the diagnosis of sarcoidosis, its diagnostic significance still remains a topic of debate. In this regard, it is worth investigating the usefulness of sACE in sarcoidosis by meta-analytic methodology. 

The present study is well designed and the data was appropriately analyzed in sufficient manner. However, there are still some concerns as follows:

1.     As the authors described in Discussion (lines 402-413), normal sACE levels depend on the deletion (D)/insertion (I) polymorphism in the ACE gene, and an ethnic factor largely influences its diagnostic ability. For instance, Song et al. reported that the prevalence of the D allele was lower in East Asian population compared to other ethnicities (J Renin Angiotensin Aldosterone Syst. 2015;16:219-26). How did the authors consider causal effects of the ACE polymorphism in the current study? The author should re-analyze and present data while considering ethnic difference in the ACE genotypes.

2.     The authors conducted meta-analysis using the published papers up to February 2022 (lines 83-84). However, more than 6 months have already passed, and there are a number of papers dealing with ACE published from March 2022 up to present. Therefore, it is better to include recent papers into the meta-analysis.

Round 2

Reviewer 1 Report

Authors significantly improved manuscript

Reviewer 2 Report

The author appropriately responded to the reviewer's comments.